# Managing Urolithiasis with Thulium Fiber Laser: Updated Real-Life Results—A Systematic Review

**DOI:** 10.3390/jcm10153390

**Published:** 2021-07-30

**Authors:** Olivier Traxer, Mariela Corrales

**Affiliations:** 1Sorbonne Université, GRC n°20, Groupe de Recherche Clinique sur la Lithiase Urinaire, Hôpital Tenon, F-75020 Paris, France; mariela_corrales_a@hotmail.com; 2Sorbonne Université, Service d’Urologie, AP-HP, Hôpital Tenon, F-75020 Paris, France

**Keywords:** laser, thulium fiber laser, holmium laser, kidney stones, ureteral stones, lithotripsy, endourology

## Abstract

Thirty-three years ago, pulsed lasers marked the beginning of a new era in endoscopic lithotripsy, and the one that was highlighted because of its potential was the Holmium: YAG laser, which became and still is the gold standard in endourology. Recently, a new laser technology has been accepted for clinical use in lithotripsy: the thulium fiber laser (TFL), showing appealing characteristics not seen before in several preclinical studies. A review of the literature was performed and all relevant in vitro studies and clinical trials until April 2021 were selected. The search came back with 27 clinical experiences (7 full-text clinical trials and 20 peer-reviewed abstracts) and 33 laboratory studies (18 full-text articles and 15 peer-reviewed abstracts). The clinical experiences confirmed the clinical safety of using the wide parameter range of the TFL. This technology demonstrated the performance at a higher ablation speed, the higher ablation efficiency, and the better dust quality of the TFL, as well as reduced stone retropulsion, thus helping to maintain an optimal visibility. No thermal or radiation damage was found. Given the current evidence, we may be facing the future gold standard laser in endoscopic lithotripsy.

## 1. Introduction

There is no doubt that the way urolithiasis is managed nowadays has changed substantially. The fundamental shift started with the introduction of light activation by the stimulated emission of radiation (LASER) technology—specifically pulsed lasers—33 years ago [1]. Pulsed lasers are the only lasers suitable for a safe endoscopic lithotripsy [1,2], and of these, the one that showed early clinical potential was the Holmium: yttrium aluminum garnet (Ho: YAG) laser [3], which is the current gold standard laser in endourology [4,5].

Recently, a new laser technology for endoscopic lithotripsies has emerged for clinical use: the thulium fiber laser (TFL) [6]. The TFL showed its appealing characteristics in several preclinical studies, possessing the widest and most flexible range of parameters among the actual laser lithotripters [7,8]. Later on, the first clinical trial was developed in Russia in 2018, making it the only country with clinical approval given by the Ministry of Health of the Russian Federation [9]. It was only in June 2020 that this technology was approved worldwide with incredible results [6]. That being said, could it be possible that the Ho: YAG era is approaching its final days?

In this paper, we aim to review the latest clinical trials available and to make a comparison between data obtained in a laboratory and real-life results.

## 2. Methods

A literature review was performed in April 2021 using the MEDLINE and Scopus databases. An additional search was performed in the medical section of the publisher Mary Ann Liebert for peer-reviewed abstract presentations that were not indexed in the previously mentioned databases. All relevant in vitro studies and clinical trials until April 2021 were selected, including case reports, case series, and conference abstracts. Exclusion criteria included the use of thulium in a non-endoscopic lithotripsy context. This review followed the Preferred Reporting Items for Systematic Reviews and Meta-Analyses (PRISMA) statement [10].

Different searches were conducted with the following Medical Subject Heading (MeSH) terms and keywords: “thulium”, “laser”, “fiber”, “lithiasis”, “kidney”, “ureter”, “lithotripsy”, “endourology”, “stones”, and “lithotripter”. Boolean operators (AND, OR) were used to refine the search. The references of each included study were also reviewed. No time period or language restrictions were applied.

The risk of bias in individual studies was assessed by the level of evidence at the study level.

## 3. Results

The PubMed and EMBASE search returned 517 articles, and 25 additional papers were added after the Mary Ann Liebert search. After duplicate removal and full review of the abstracts and texts gathered, a total of 60 full-text articles and peer-reviewed abstract presentations were included for qualitative analysis, including 27 clinical experiences (7 full-text clinical trials and 20 peer-reviewed abstracts) and 33 laboratory studies (18 full-text articles and 15 peer-reviewed abstracts). The summary of the selection process is represented in Figure 1.

For better meeting the main purpose of the present report, the laboratory results obtained were divided into different sections according to their main purpose and compared to the clinical results obtained with TFL in real life (Table 1). Those sections are organized as follows: laser settings, laser fibers, TFL outcomes (ablation efficiency, speed, and operation time), dust quality, retropulsion and visibility, temperature safety, radiation and electrical safety, and finally, SFR. Additionally, the overall results of the seven recent full-text clinical trials are summarized in Table 2.

### 3.1. Laser Settings

The TFL differs from the Ho: YAG laser in many aspects. Instead of using a flash lamp as the energy source, the TFL uses several laser diodes that excite the thulium ions contained inside a very long and thin fiber (10–20 μm core diameter), and the emitted laser beam (wavelength of 1940 nm) can work in a continuous or pulsed mode with the widest range of parameters ever seen in the market [11,12]. The pulse energy can go as low as 0.025 joules (J) and as high as 6 J, and the pulse frequency can reach up to 2400 hertz (Hz), with a peak power of 500 watts (W) and an average power of 2–60 W. Additionally, we can choose between a short or a long pulse duration (200 µs–50 ms) [8,11,12,13,14]. These characteristics largely surpass those of the Ho: YAG technology. The least energy that the Ho: YAG laser can provide is 0.2 J, and the highest frequency and power that it can reach in recent new devices (MOSES™ 2.0) are 120 Hz and 120 W, respectively [15,16]. As mentioned, TFL technology offers greater versatility and control of pulse parameters than the Ho: YAG laser does [11,14,17,18].

The initial clinical experiences have demonstrated that the different laser settings used in the operating room are safe for patients. Many combinations of laser settings have been tested for ureteral, renal, and bladder stone lithotripsies presented in various conference abstracts [9,19,20,21,22,23,24,25,26,27,28,29,30,31,32,33,34,35,36] and in full-text clinical experiences [37,38,39,40,41,42,43] (Table 2). However, the ideal one has not yet been established.

### 3.2. Laser Fibers

One of the most important factors to evaluate before using a new laser fiber is the risk for optical fiber fractures at the moment of deflection. It has been demonstrated that TFL laser fibers are more capable of resisting important ureteroscope deflections than the Ho: YAG laser is when using the same 200 μm core-diameter laser fiber (CDF) at the same laser settings [44]. Even when smaller TFL laser fibers are used, such as 50 or 150 μm CDF, the energy delivered under extreme bending configurations is optimal to perform an adequate lithotripsy, being compatible with lower pulse energies and higher pulse rates [45,46]. Furthermore, one of the advantages of the 50 μm CDF is the enhanced irrigation and the fact that it requires 30 times less the cross-sectional area of the standard 270 μm CDF [45]. Knudsen et al. demonstrated that 150 and 200 μm TFL CDFs are more flexible and able to bend to a smaller diameter than the Accumax 200 and MOSES^TM^ 200 fibers are [47]; furthermore, the thulium fiber system offers less fiber burnback than the 120 W Ho: YAG laser does [48].

One possible explanation for this bending resistance and excellent lithotripsy performance, even with extreme deflection, may reside in the Gaussian spatial beam profile emitted from the 18-micrometer-core thulium-doped silica fiber, which may be focused to spot diameters comparable to tens of micrometers, allowing the TFL itself to couple into small surgical fibers [45]. This spatial beam profile is much thinner and more uniform and symmetrical than the one produced by the Ho: YAG laser [44,45].

Initial clinical experiences have confirmed that the use of a small TFL laser fiber gives a better instrument deflection, and therefore, the urologist is able to treat lower pole calyceal stones [21,25], even with an acute lower pole infundibulopelvic angle [21].

### 3.3. TFL Outcomes

The ablation speed of the TFL is faster than the one seen with the Ho: YAG laser at any setting [8,18,49,50,51,52,53,54]. Comparing both laser technologies under identical parameters, it has been shown that the ablation speed of the TFL is up to 2 to 5 times higher in the fragmentation and dusting modes, respectively [49,52,53], independently of the stone composition analyzed in the laboratory, with calcium-oxalate monohydrate or uric acid stones (human stones or artificial ones) being the most analyzed [8,49,50]. Furthermore, comparisons have also been made with high-power Ho: YAG lasers (120 W) offering the Moses technology (MT), and the results show that the TFL is significantly faster with [55] or without the MT [50]. These results could be explained by the higher ablation rates obtained with the small CDF of the TFL, regardless of the laser settings [49] and due in part to the combination of the TFL’s high pulse rate, high average power, and lower stone retropulsion [18]. In terms of fiber laser diameter, one report mentions that both TFL laser fibers, 150 and 272 CDF, were related with a twofold and threefold higher ablation rate compared to the Ho: YAG laser fiber of 272 CDF in “dusting” mode [51]. The average ablation speed of renal stones is 0.7 mm^3^/s when the lithotripsy is performed by a standard low-power Ho: YAG laser (30 W) [56].

Recent clinical trials have corroborated the higher ablation speed of the TFL, which performs at least two times faster than the current gold standard laser [37,38,39,40,41,42,43] (Table 2). Even compared to high-power Ho: YAG lasers (up to 120 W and 80 Hz), the TFL is able to deliver a faster ablation speed by a factor of 1.5 and 3 in the fragmentation and dusting modes, respectively [22].

On the other hand, in terms of ablation efficiency (J/mm^3^), also known as the total energy needed to ablate 1 mm^3^ of stone volume [56], in vitro/ex vivo analyses of this new technology have shown promising results in ablating all types of stones [57,58]. The Ho: YAG laser (30 W) has a median of 19 J/mm^3^ for stones with median Hounsfield units (HU) of 1040, requiring more energy if the stone density is over 1000 HU [56]. Compared to that laser, the volume ablated by a single pulse with a TFL is at least three times higher [52,59]. With equal settings and laser fibers (272 μm CDF), ablation volumes are up to four- and twofold higher with the TFL than with the Ho: YAG laser in dusting and fragmentation modes, respectively, using “soft” (uric-acid-like) and “hard” (calcium-oxalate monohydrate-like) stones [51]. Furthermore, the 150 μm CDF of the TFL results in ablation volumes that are still fifty percent higher than those of a Ho: YAG laser with 272 μm CDF [51]. Comparisons have also been made with high-power Ho: YAG lasers (120 W) with MT, and TFL resulted more efficacious when maximal dusting settings were used [54,60]. Using various Ho: YAG pulse delivery modes (short pulse, long pulse, and Moses pulse), Ventimiglia et al. mentioned that the ablation results indicated that both the increase in pulse width and the decrease in peak power provide about 1.5–2 times higher ablation efficiency [17].

Current clinical full-text evidence confirms that less energy is needed to ablate 1 mm^3^ of stone volume, regardless of the stone density [37,38,39,40,41,42,43] (Table 2), which agrees with the clinical experience presented in different conference abstracts about the TFL’s capacity to ablate all stone types [9,20,21,22,25,26,28,31,32,33,34,35]. This more efficient energy delivery can be hypothesized to be due to two factors: the water absorption and the pulse profile. The TFL’s higher water absorption at 1940 nm (about 4–5 times higher than the one observed with the Ho: YAG at 2100 nm) [61], translating into a higher water absorption of the laser energy by the water contained within pores near the stone surface, could play a critical role in stone ablation [17]. Concerning the pulse profile, the fact of being uniform in shape (square pulse shape), unlike the holmium laser, allows the energy to be more evenly delivered to the stone during the ablation process [59]. Controversies are still surrounding the pulse duration. It is less clear whether the pulse duration plays a role or not in the ablation efficiency; however, the TFL has the advantage of being able to work with both low pulse energies and a longer pulse duration (up to five times longer than that of the Ho: YAG laser), characteristics that are desired during stone dusting to reduce retropulsion [50,59] and to achieve an efficient lithotripsy.

### 3.4. Dust Quality

To date, there is no consensus regarding the exact definition of dust; nonetheless, a size limit of ≤250 μm seems to properly adhere to an adequate definition of stone dust capable of being aspirated through the working channel [62]. Laboratory studies have suggested that the TFL produces smaller fragments than the Ho: YAG laser does in dusting mode, regardless of the stone type or laser settings [50,63]. It is also remarkable that the TFL is capable of producing twice as much dust as the Ho: YAG laser with MT [55,64].

Recent clinical trials have corroborated the smaller fragments produced with the thulium technology [37,38,39,40,41]. The smaller fragments are responsible for the birth of a new size-related definition of stone dust, namely micro-dust, a term recently proposed to better define stone particles smaller than the 150 μm CDF of the TFL [39].

### 3.5. Retropulsion and Visibility

In the laboratory, results mentioning the lower retropulsion given by the TFL are encouraging. One report concluded that it seems that the effect of retropulsion from one thulium laser pulse (radiation energy 3 J) is 3 mm, which is three times less than that from one holmium laser pulse (10.5 mm) [49]. Various other studies have demonstrated that the TFL causes slower retropulsion when compared to Ho: YAG for equivalent energy, frequency, and average power settings [8,18,50,53], irrespective of the mode selected (short pulse, long pulse, or Moses pulse) of the Ho: YAG laser device [8,17,48].

In real life, urologists that have tested the thulium technology in lithotripsy have corroborated the insignificant retropulsion under different laser settings, mostly when using energy levels below 0.5 J [9,20,26,31,35]. Most recent publications have used other laser parameters (including >0.5 J), and the results are consistent [39,40,41,42,43]. Additionally, when comparing both technologies, the stone retropulsion level of the TFL was greatly reduced [22,38]. It is important to remark that most clinical experiences previously mentioned have found the TFL to have an optimal visibility [9,20,26,31,35,39,40,41,42,43], even when higher frequency regimens were applied (0.15 J × 200 Hz) [41].

A possible explanation of the lower retropulsion of the TFL is probably because of the more rectangular pulse, smaller optical fibers, and the combination of lower peak power with lower pulse energies, without sacrificing ablation efficiency [17,18,38]. It is well known that lower peak power produces lower pressure in the laser-induced bubble and a smaller water stream [10], resulting in reduced stone displacement.

### 3.6. Operating Time

Based on the technological superiority of the TFL shown in the laboratory, prior to clinical experiences, the suggestion of reducing the operating time was proposed [65,66].

One of the most recent clinical experiences by Martov et al. comparing the Ho: YAG laser and the TFL found a statistically significant shorter total operation and laser time with the latter technology [38].

The previously mentioned advantages of the TFL, such as the widest range of parameters, higher ablation speed, higher ablation efficiency, and less stone retropulsion, seem to lead to significant time-saving.

Further studies comparing both laser technologies in real-life scenarios need to be performed.

### 3.7. Temperature Safety

This is still a subject of controversy. Due to the higher water absorption of the TFL, it has been considered that it may lead to a higher temperature rise [67] and, thus, thermal damage (temperatures over 54 °C) [68,69,70]. However, comparative studies were performed between the TFL and high-power Ho: YAG laser devices, and it was not only observed that both lasers had a similar temperature increase under the same laser settings (dusting, pop-dusting, fragmenting, or pop-corning) [8,70,71], but also that when cooler fluids were applied, the time needed to reach a dangerous temperature (>45 °C) was longer [70,71]. Both technologies need the same precautions. As long as there is moderate irrigation, TFL is safe to use [18,72,73]. However, as irrigation rates decrease, even lower power settings can produce a significant temperature increase, potentially leading to urothelial tissue injuries [73]. Recent ex vivo studies showed that higher power settings with both high-power Ho: YAG laser and TFL cause a higher temperature rise in the ureter during lasering—a temperature rise that is equivalent during dusting but higher during TFL fragmentation; nonetheless, neither laser reached the threshold for thermal injury [74].

To date, no current clinical experiences have shown evidence of severe tissue thermal damage during surgery or follow-up [9,20,21,22,25,26,28,31,32,33,34,35,37,38,39,40,41,42,43].

### 3.8. Radiation Safety

Eye injury is one of the main concerns of the urologist when performing a laser surgery. It is known that when lasers emit wavelengths longer than 1400 nm, they are considered to be “eye-safe” because most of the radiation is absorbed by the cornea [12,75]. As the TFL wavelength is 1940 nm, we can say that this laser fits into this category. Until now, no retina damage has been reported in clinical experiences.

### 3.9. SFR

Stone-free status was defined as the absence of any residual fragments > 2 mm assessed by computed tomography (CT) performed at 1 month [3] or at 3 months after surgery, using either a non-contrast low-dose technique [3,40,43] or a contrast-enhanced one [41,42] with the objective of assessing short-term complications concomitantly (i.e., strictures). One study only used kidney, ureter, and bladder imaging (KUB X-ray) coupled to CT as an SFR parameter at one month after surgery [37].

Focusing on the most recent clinical experiences [37,38,40,41,42,43], we can say that the SFR is optimal for this new technology, as expected after various in vitro/ex vivo studies, ranging from 85 to 100% (Table 2). Further clinical trials with longer-term follow-ups are necessary.
jcm-10-03390-t001_Table 1Table 1Comparison between TFL preclinical and clinical studies.In Vitro/In Vivo Clinical ExperienceLaser settings: Pulse frequency, energy, and total power-Significantly longer pulses and lower peak power than those generated by the Ho: YAG laser at similar energy settings [8,11,12,13,14,15,16]-Greater versatility and control of pulse parameters than the Ho: YAG laser [11,14,17,18]-Confirmed with safety [9,19,20,21,22,23,24,25,26,27,28,29,30,31,32,33,34,35,36,37,38,39,40,41,42,43]Laser fibers-More flexible [44,45,46] and able to bend to a smaller diameter than the Accumax 200 and MOSES^TM^ 200 fibers [47]-Less fiber burnback in comparison to the 120 W Ho: YAG laser [48]-Able to reach difficult anatomical positions [21,25]Ablation speed and ablation efficiency-Ablation speed: at least 2 times faster than the Ho: YAG laser, at any setting [8,18,49,50,51,52,53,54], including MT [55]-Ablation efficiency: up to 2 times higher [17,51,52,54,59,60], capable of lasering all type of stones [57,58]-Ablation speed: confirmed [22,37,38,39,40,41,42,43]-Ablation efficiency: confirmed [37,38,39,40,41,42,43], regardless of the stone density [9,20,21,22,25,26,28,31,32,33,34,35,37,38,39,40,41,42,43]Dust quality-Smaller dust than the Ho: YAG laser [50,63]-Produces twice as much dust as the Ho: YAG laser with MT [55,64]-Smaller dust: confirmed [37,38,39,40,41]Retropulsion and visibility-Less retropulsion in comparison to the Ho: YAG laser [8,17,18,48,49,50,53]-Confirmed in comparative studies [22,38]-Minimal retropulsion in non-comparative studies [9,20,26,31,35,39,40,41,42,43]-Optimal visibility [9,20,26,31,35,39,40,41,42,43]Operating time-Reduction in the operation time [65,66]-Confirmed [38]Temperature safety-No significant difference in water temperature elevation between TFL and Ho: YAG laser [8,70,71,74]-TFL and Ho: YAG laser share similar risk profile and irrigation precautions [18,72,73,74]-No evidence of severe soft tissue damage in the current clinical experiences [9,20,21,22,25,26,28,31,32,33,34,35,37,38,39,40,41,42,43]Radiation safety-Shorter optical penetration depth [12,75]-Lacking long-term evidenceTFL: Thulium fiber laser; MT: Moses technology.
jcm-10-03390-t002_Table 2Table 2Full-text TFL clinical experiences.ReferenceNProcedureGroupingAge (years)Stone Size StoneDensity (HU)Operating Time (min)LaserSettingsLOT Ablation Speed (mm^3^/s)Ablation Efficacy(J/mm^3^)SFR (%)Enikeev et al., 2020 [42] PCS 120PCNL
52 ± 1.812.5 ± 8.8 mm1019 ± 37523.4 ± 17.90.8 J31–38 Hz25–30 W5 ± 5.7 minN. A
85Enikeev et al., 2020 [41] PCS40RIRS(Renal stones)15 HF25 HRF56883 (606–1664) mm^3^880 ± 38123.1 ± 10.90.5 J × 30 Hz 15 W0.15 J × 200 Hz30 W219 (90–330) s372 (96–414) s5.5 (1.5–8.7)8.5(3.6–19)2.7(1.8–9.8)4.8(2.6–11.3)92.5Martov et al., 2020 [3] RCT174RIRS(Ureteral stones)87 TFL87 HL
12.2 ± 0.1 mm11.3 ± 0.1 mm1001 ± 266994 ± 21424.7 ± 0.732.4 ± 0.71 J × 10 Hz10 W1 J × 10 Hz10 W8.4 ± 0.4 min15.9 ± 0.5 minN. AN. A10094Shah et al., 2020 [37] PCS54RIRS(Renal stones)
40.42 ± 15.172337.75 ± 1996.84 mm^3^1300.55 ± 435.3239.85 ± 20.520.1–1 J × 100–300 Hz605.37 ± 464.51 s5.02 ± 3.93N. A100Corrales et al., 2021 [39] PCS50 RIRS9 Ureteral stones41 Renal stones66(55.5–74)55(44–61.5)486(332–1250) mm^3^1800(682.8–2760) mm^3^998(776–1300)1200(750–1300)N. A0.2–0.4 J × 20–55 Hz6.5–16 W0.2–0.6 J × 50–180 Hz20–32 W9.3(7.3–17) min23(14.2–38.7) min0.7(0.3–1.6)1.16(0.8–2.1)16.3(8.6–35.5)18.6(9.5–26.1)N. AKorolev et al., 2021 [40] PCS125Mini–PCNL36 LF75 HF14 HRF 52 ± 1.82386 (1083–4202) mm^3^1186 (905–2317) mm^3^1337 (878–3665) mm^3^900(625–1275)1100(750–1350)1170(636–1300)10.15(3.9–13.25)17(5.8–27.5)22(8.25–29)0.5–6 J × 4–19 Hz8–50 W0.5–2 J × 20–49 Hz10–40 W0.1–0.5 J × 50–200 Hz12–30 W319(198–576) s312(177–528) s354(213–632) s6.8(4.6–12.5)5.1(3–8.7)4.4(3.4–7.6)3.38(1.68–5.36)4.89(2.77–7.44)4.21(3.3–6.12)85%Enikeev et al., 2021 [43] PCS149RIRS(ureteral stones)
N. A179(94–357) mm^3^985 ± 360<600.1–1 J × 7.5–200 Hz1.2(0.5–2.7) min140(80–279)5.6(3–9.9)90%PCS: Prospective clinical study; RCT: Randomized control trial; N: Sample size; N.A: Not available; PCNL: Percutaneous nephrolithotomy; RIRS: Retrograde intrarenal surgery; LF: Low frequency; HF: High frequency; HRF: Higher frequency; LOT: Laser-on time; TFL: Thulium fiber laser; HL: Ho: YAG Laser; LP: Lower pole; MP: Mid-pole; UP: Upper pole.

## 4. Limitations

As TFL is a new technology recently made available worldwide, there are only a few studies reporting the TFL’s initial clinical results. To date, the TFL does not seem to have any downsides when taking the same precautions as for the Ho: YAG laser. However, most studies included in this review, analyzing the advantages of TFL alone or over the Ho: YAG laser, were in vitro and in vivo studies. Unfortunately, only one clinical RCT comparing both technologies is available in the literature [3]. To avoid future bias, such as selection or performance bias, it is necessary to conduct further comparative and prospective RCTs as well as a long-term follow-up to corroborate what we are witnessing with the TFL.

On the other hand, the results shown are exciting, and perhaps we are starting a lasting laser transition. The objective of this review was to analyze all the TFL-related clinical evidence, mainly with the most recent information published (seven full-text clinical experiences), comparing the current clinical results with those obtained in the laboratory.

## 5. Conclusions

The latest publications have concluded that the TFL is safe in endoscopic lithotripsy. All the speculations about this novel technology generated by laboratory trials are starting to be confirmed, and this promising technology may become the new gold standard in the near future.

## Figures and Tables

**Figure 1 jcm-10-03390-f001:**
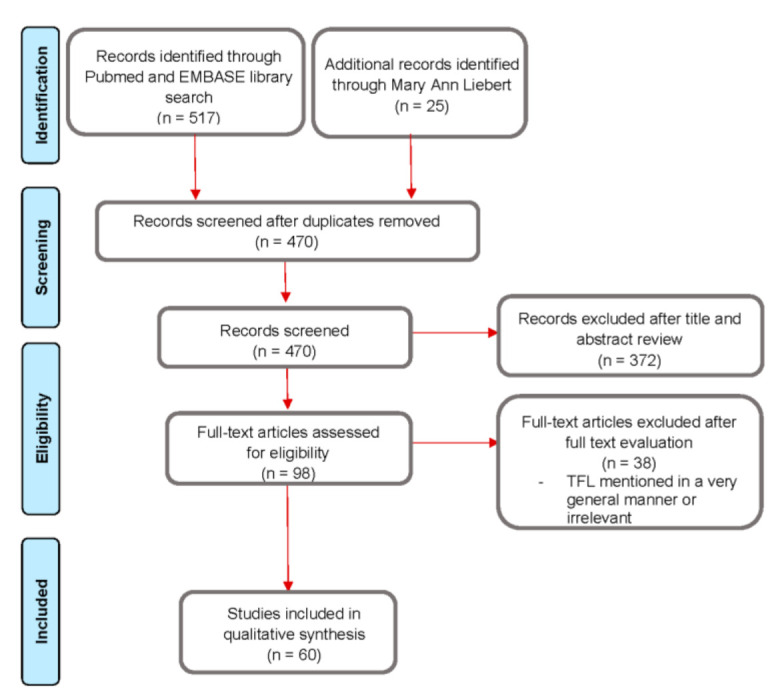
Flow chart of the literature review.

## Data Availability

Data are available by contacting authors.

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
