# Peer review of "Managing Urolithiasis with Thulium Fiber Laser: Updated Real-Life Results—A Systematic Review"

_jcm, 2021, doi:10.3390/jcm10153390_

Round 1

Reviewer 1 Report

Congratulations on your work and efforts to summarize this exciting and emerging technology. 

This review certainly incorporates the many studies available in a consistent manner.

From a methodological standpoint, this manuscript mirrors prior work (PMID: 31656746). While for the vast majority of studies it may not be beneficial, for the major studies described perhaps assessing study-level or outcome level risk of bias may help with the generalization of the conclusions.

Additionally, it may be of interest to include how SFR was assessed in the studies described.

This review describes the advantages of TFL very well. The limitations of TFL are not as well described. Are there no downsides to the TFL? If there is no published work describing potential downsides to the TFL, it would be helpful to describe that in the manuscript.

It is clear that prior work describes improved dusting and smaller fragments as a result of TFL lithotripsy. An interesting area to perhaps expand upon is the lasers ability to fragment, if desired. A prior study by Hardy et al. Analysis of thulium fiber laser induced bubble dynamics for ablation of kidney stones, which showed TFL produces a smaller bubble profile with lower collapse pressure than the Ho:YAG laser pulse. Does this laser characteristic or perhaps its lower peak energy decrease the fragmentation or pop-corning ability of the TFL. This may be of particular interest to urologists who prefer fragmentation and basket extraction techniques, as the ability to create larger fragments may increase efficiency.

Author Response

Reviewer 1:

Congratulations on your work and efforts to summarize this exciting and emerging technology.

This review certainly incorporates the many studies available in a consistent manner.

Answer: We are thankful for a very positive comment from the reviewer about our paper.

From a methodological standpoint, this manuscript mirrors prior work (PMID: 31656746). While for the vast majority of studies it may not be beneficial, for the major studies described perhaps assessing study-level or outcome level risk of bias may help with the generalization of the conclusions.

Answer: PMID: 31656746 was basically a systematic review that aimed to evaluate the reality and expectations of the new TFL lithotripter. However, clinical trials were not taken into account due to the laser novelty. We agree with the reviewer about assessing the study-level or outcome level risk of bias for the major studies described. All changes are highlighted in red. We have now added this paragraph at the end of Methods, which reads –

‘The risk of bias in individual studies was assessed by level of evidence at the study level’.

Additionally, it may be of interest to include how SFR was assessed in the studies described.

Answer: We strongly agree with your comment. We have now added this paragraph at the beginning of the SFR subtitle, which reads –

‘Stone free status was defined, mostly, as the absence of any residual fragments > 2 mm assessed by computed tomography (CT) performed at 1 month [3] or at 3 months after surgery, using whether a non-contrast low dose technique [3, 40, 43] or a contrast-enhanced one [41, 42] with the objective of assessing short-term complications concomitantly (i.e strictures). One study only used as a SFR parameter the kidneys, ureters and bladder imaging (KUB X-ray) coupled to the CT at one month after surgery [37]’.

This review describes the advantages of TFL very well. The limitations of TFL are not as well described. Are there no downsides to the TFL? If there is no published work describing potential downsides to the TFL, it would be helpful to describe that in the manuscript.

Answer: We support your comment. In fact, apparently there are no downsides when using properly the TFL, taking into count that the operator has to follow the same precaution measures that the ones used with the Ho:YAG laser. We have now modified the “Limitations” part, which reads –

‘TFL is a new technology recently available worldwide, there are small studies on-ly reporting TFL’s initial clinical experience. Up to date, TFL does not seem to have any downside, when taking the same precautions as for the Ho:YAG laser. However, most studies included in this review, analyzing the advantages of TFL alone or over the Ho:YAG laser, are in/vitro and in vivo. Unfortunately, only one clinical RCT com-paring both technologies is available in the literature [3]. To avoid future bias, such as selection and performance bias, it is mandatory to have further comparative and pro-spective RCT, as well as a long-term follow up, to corroborate what we are witnessing with the TFL’.

It is clear that prior work describes improved dusting and smaller fragments as a result of TFL lithotripsy. An interesting area to perhaps expand upon is the lasers ability to fragment, if desired. A prior study by Hardy et al. Analysis of thulium fiber laser induced bubble dynamics for ablation of kidney stones, which showed TFL produces a smaller bubble profile with lower collapse pressure than the Ho:YAG laser pulse. Does this laser characteristic or perhaps its lower peak energy decrease the fragmentation or pop-corning ability of the TFL. This may be of particular interest to urologists who prefer fragmentation and basket extraction techniques, as the ability to create larger fragments may increase efficiency.

Answer: We thank the reviewer for the accurate comment. It is true that because the TFL has a lower peak power (up to 500 W), the stone displacement is reduced. We have clarified that point in “Retropulsion and visibility” adding a paragraph which reads –

‘It is well-known that lower peak power produces lower pressure in the laser-induced bubble and a smaller water stream [10], resulting in reduced stone displacement’.

The fact that the TFL have this low peak power does not reduce its capability to perform a proper fragmentation or pop-corning. In the same “Retropulsion and visibility” part, we specify that “The possible explanation of the lower retropulsion of the TFL is probably because of the more rectangular pulse, smaller optical fibers and the combination of lower peak power with lower pulse energies, without sacrificing ablation efficiency [17,18,38]”. Additionally, we remark in the second paragraph of the “TFL outcomes” part that with fragmentation settings, the TFL ablates 2 times faster than the Ho:YAG laser: “With equal settings and laser fibers (272 μm CDF), ablation volumes are up to four and two-fold higher with TFL than with the Ho: YAG laser in dusting and fragmentation mode, respectively, using “soft” (uric acid like) and “hard” (calcium-oxalate monohy-drate like) stones [51]”.

Reviewer 2 Report

Dear Editor,

 This article reviews in vitro studies and clinical trials until April 2021 on managing urolithiasis with Thulium fiber laser. However, similar reviews have recently been published with a wide overlap in reviewed literature [1, 2].

Additionally, when excluding two product data sheets from the reference list, 28 of 73 references (38%) have one of the authors of the manuscript handed in in their author list as well. Overall, there seem to be a limited number of groups having performed studies with the thulium laser so far – 40 of 73 (55%) of the references include on or more of the following persons: Martov, A., Dymov, A., Enikeev, D., Traxer, O.

In my opinion, the paper lacks in clarity and in depth at some points (see comments below).

Taking the last two points into account, I think the revision and inclusion of further studies then conducted would be better.

Comments

Methods

  • The keywords used for the literature search are more or less the ones given in [1]

Results

  • This may be due to editing, but the tables are not clearly arranged.

Laser fibers

I think this paragraph is highly misleading: the treatment fibers used for holmium as well as thulium fiber lasers are quartz fibers with a low content of hydroxyl ions (and usually a numerical aperture of 0.22). The main point is given in paragraph 2 of this section: the differences in the beam profile of a holmium and a thulium fiber laser and how they can be coupled into a fiber (I would not call this ‘radiation profile’, and ‘better’ is not giving enough information on how the beam profile is different to that of a holmium laser).

TFL outcomes

  • Also in this section, ‘TFL laser fibers’ and Ho:YAG laser fibers’ are mentioned. Please see comment above.

Retropulsion and visibility

  • ‘On report concluded that it seems that the effect of retropulsion from one thulium laser pulse (radiation energy 3J) is 3mm ...’ ß this is cut down too far – important information is missing

[1]        Kronenberg, P., Hameed, B. Z., & Somani, B. (2021). Outcomes of thulium fibre laser for treatment of urinary tract stones: results of a systematic review. Current Opinion in Urology, 31(2), 80.

[2]        Khusid, J. A., Khargi, R., Seiden, B., Sadiq, A. S., Atallah, W. M., & Gupta, M. (2021). Thulium fiber laser utilization in urological surgery: A narrative review. Investigative and Clinical Urology, 62(2), 136.

Author Response

Reviewer 2:

We thank the reviewer for its commentary. Indeed, clinical data about this new technology is missing. We agree that similar publications have been made about this subject [1] and [2] (cited by the reviewer); but we have added new and recent clinical data that was not taken into consideration in those previously studies, focusing on stone management and not in all TFL advantages as in the prostate field [2].

All changes are highlighted in red.

Methods

The keywords used for the literature search are more or less the ones given in [1]

Answer: Perhaps the keywords seem similar because of the subject in question, but they are not totally the same. We are sorry if it looks alike but are the keywords we used.

Results

This may be due to editing, but the tables are not clearly arranged.   

Answer: Thank you for that remark, we also agree that the tables are not clearly arranged. We believe is an editing problem because we send the tables with a proper format. We tried to format the tables and place them almost at the end of the article.

Laser fibers

I think this paragraph is highly misleading: the treatment fibers used for holmium as well as thulium fiber lasers are quartz fibers with a low content of hydroxyl ions (and usually a numerical aperture of 0.22). The main point is given in paragraph 2 of this section: the differences in the beam profile of a holmium and a thulium fiber laser and how they can be coupled into a fiber (I would not call this ‘radiation profile’, and ‘better’ is not giving enough information on how the beam profile is different to that of a holmium laser).

Answer: We thank the reviewer for the accurate comment. We have now given a more detailed explanation in the second paragraph of “Laser fibers” which reads –

‘One possible explanation for this bending resistance and excellent lithotripsy per-formance even in an extreme deflection, may reside in the Gaussian spatial beam profile emitted from the 18-μm-core thulium-doped silica fiber, which may be focused to spot diameters comparable to tens of micrometers, allowing the TFL itself to couple into small surgical fibers [45]. This spatial beam profile is much thinner, uniform and sym-metrical than the one produced by the Ho:YAG laser [44, 45]’.

TFL outcomes

Also in this section, ‘TFL laser fibers’ and Ho:YAG laser fibers’ are mentioned. Please see comment above.

Answer: We have noticed your remark and now, we have made the modifications, as you mentioned above.

Retropulsion and visibility

‘On report concluded that it seems that the effect of retropulsion from one thulium laser pulse (radiation energy 3J) is 3mm ...’ ß this is cut down too far – important information is missing

Answer: This information is taken from an in vitro study [49], that was the result that the authors obtained (one of our 15 peer-reviewed abstracts). Unfortunately, there are no more information on that abstract. However, later on, we explain similar results achieved in other experiences, in the laboratory and in the real life.

Reviewer 3 Report

Thank you for submission of the review for publication of JCM. This is not an original article but a review article. In this case, the author described the systematic review method.

Beyound prostate, Holmium is the most popular in most institute for lithotripsy. Despite of review, lack of RCT and comparative study with Holmium are the limitations. This review is worthy to give information of Thulium laser for lithotripsy.

Author Response

Reviewer 3:

Thank you for submission of the review for publication of JCM. This is not an original article but a review article. In this case, the author described the systematic review method.

Beyound prostate, Holmium is the most popular in most institute for lithotripsy. Despite of review, lack of RCT and comparative study with Holmium are the limitations. This review is worthy to give information of Thulium laser for lithotripsy.

Answer: We are thankful for the positive comment from the reviewer about our paper and we are aware of the limitations, the lack of RCT and comparative studies with Ho:YAG laser.

Reviewer 4 Report

The paper is well written, interesting and discuss a topical issue in the field of stone treatment. 

However, a greater caution is needed in the data presentation and in the discussion.

It should be clarified that all the comparative studies between Ho: YAG laser and TFL described in the paper and supporting the advantages of Thulium are in vitro/in vivo. Only one clinical RCT compared in the really clinical practice the two techniques (Martov et al.). The other 6 peer reviewed papers available in literature are able tu confirm the safety and efficacy of the TFL, but do not demonstrate a superiority compared to Ho: YAG laser.

Moreover, the majority of the study supporting the role of TFL are abstract, which were not published in full, thus preventing to assess any study biases.

I would kindly suggest  that the authors amend the discussion and the limitations chapter on the basis of these considerations.

Author Response

Reviewer 4

The paper is well written, interesting and discuss a topical issue in the field of stone treatment.

Answer: We are thankful for a very positive comment from the reviewer about our paper.

However, a greater caution is needed in the data presentation and in the discussion.

It should be clarified that all the comparative studies between Ho: YAG laser and TFL described in the paper and supporting the advantages of Thulium are in vitro/in vivo.

Only one clinical RCT compared in the really clinical practice the two techniques (Martov et al.). The other 6 peer reviewed papers available in literature are able tu confirm the safety and efficacy of the TFL, but do not demonstrate a superiority compared to Ho: YAG laser.

Answer: We thank the reviewer for the accurate comment. We have now emphasized that remark by separating the clinical experiences from the laboratory ones. In each subtitle you will find first, the laboratory experience and in a separate paragraph, the clinical experience (that order only differs in the “SFR” part). We agree that it should be mentioned that further comparative RCT are needed, that is why we have reformulated the first part of “Limitations” which reads –

‘TFL is a new technology recently available worldwide, there are small studies on-ly reporting TFL’s initial clinical experience. Up to date, TFL does not seem to have any downside, when taking the same precautions as for the Ho:YAG laser. However, most studies included in this review, analyzing the advantages of TFL alone or over the Ho:YAG laser, are in/vitro and in vivo. Unfortunately, only one clinical RCT com-paring both technologies is available in the literature [3]’.

Moreover, the majority of the study supporting the role of TFL are abstract, which were not published in full, thus preventing to assess any study biases.

I would kindly suggest that the authors amend the discussion and the limitations chapter on the basis of these considerations.

Answer: We thank the reviewer for that remark. We have now added a paragraph about the possible bias of a non RCT, which reads –

‘To avoid future bias, such as selection and performance bias, it is mandatory to have further comparative and prospective RCT, as well as a long-term follow up, to corroborate what we are witnessing with the TFL’

Round 2

Reviewer 4 Report

The paper is now suitable for publication